# Comparison of laparoscopic tubal-preserving surgical management versus methotrexate therapy for tubal pregnancy: A conventional and network meta-analysis

**Junwei Liao, Yong Lin** 👤*, **Yan Long**

Luzhou Maternal And Child Health Hospital (Luzhou Second People's Hospital), Gynecology, Luzhou City, Sichuan Province, China

* 2043359232@qq.com

## Abstract

### Background

Tubal pregnancy is one of the common emergencies in obstetrics and gynecology. There are various treatment options for tubal pregnancy. However, there is no consensus on how patients should choose among the treatment options that preserve the fallopian tube. This study aims to investigate the difference in prognosis between different treatment options when the tube is preserved in patients with tubal pregnancy.

### Methods

We conducted a search for studies on tubal pregnancy in the PubMed, EMBASE, Web of Science, and Cochrane Library databases. Ultimately, we obtained 8 studies that met the screening criteria. The Jadad scale and NOS were used to evaluate the quality of the studies, and the evidence quality was evaluated and graded according to GRADE guidelines. Stata 17.0 software was used for data analysis.

### Result

A total of 8 studies involving 677 patients were included. Meta-analysis revealed a significant overall difference in the failure rate between methotrexate treatment and salpingostomy for tubal pregnancy(OR=1.582, 95%CI(1.062, 2.358), p = 0.024). Subgroup analysis demonstrated that a single dose of methotrexate was associated with a higher failure rate compared to salpingostomy(OR = 2.044,95%CI(1.20, 3.47), p = 0.008). In contrast, two or more doses of methotrexate did not show a significant difference in failure rate compared to salpingostomy(OR = 1.130, 95% CI(0.62, 2.07), p = 0.692). However, salpingostomy combined with methotrexate showed a lower failure rate compared to salpingostomy alone(OR = 0.11, 95% CI (0.03–0.48), p = 0.003).

**Data availability statement:** All relevant data are within the manuscript and its Supporting Information files.

**Funding:** The author(s) received no specific funding for this work.

**Competing interests:** The authors claim there is no conflict of interest.

Network meta-analysis indicated that the salpingostomy combined with methotrexate had the lowest failure rate, while there was no significant difference in failure rates between salpingostomy and methotrexate alone.

## Conclusion

For hemodynamically stable patients with a desire for future fertility, salpingostomy combined with methotrexate is an excellent option for treating tubal pregnancy. This approach has a lower failure rate compared to either methotrexate or salpingostomy alone and appears to have minimal impact on the patient's quality of life.

## 1 Background

Ectopic pregnancy (EP) is defined as a pregnancy occurring outside the uterine cavity, most commonly in the fallopian tubes (96%) [1]. Tubal pregnancy, accounting for over 95% of ectopic pregnancies, refers to the implantation of a fertilized ovum within the fallopian tube [2]. Under normal physiological conditions, the fertilized ovum traverses the fallopian tube and implants in the endometrial lining. However, in tubal pregnancies, impaired tubal function—caused by factors such as inflammation, adhesions, surgical history, or congenital anomalies—prevents the embryo from migrating to the uterus, resulting in ectopic implantation.

The incidence of tubal pregnancy has been rising globally, accounting for approximately 1–2% of all pregnancies [3]. As a common obstetric emergency, untreated tubal pregnancies may lead to fallopian tube rupture, massive hemorrhage, and life-threatening complications [4]. Current treatment modalities include both conservative and surgical approaches, each associated with varying rates of subsequent intrauterine pregnancy (IUP) and recurrent ectopic pregnancy (REP) depending on the chosen intervention [5].

Treatment options for tubal pregnancy vary and include expectant management, methotrexate (MTX) therapy, salpingectomy, and salpingostomy. A prospective study demonstrated higher(OR = 1.61, 95% CI: 1.29–2.01) postoperative spontaneous pregnancy rates in patients undergoing salpingostomy compared to those who underwent salpingectomy [6], underscoring its advantage in preserving reproductive potential. Studies indicate that MTX treatment does not achieve a 100% success rate in hemodynamically stable EP patients [7–9]. Similarly, salpingostomy carries a risk of retained trophoblastic tissue, which may necessitate adjuvant medical therapy or repeat surgical intervention [10,11]. While growing attention has been directed toward reproductive outcomes in EP patients, particularly those desiring future fertility, treatment failure rates remain a critical concern due to their direct impact on patient recovery time and healthcare costs [12–15].

Therefore, *it is important to clarify* the failure rates associated with different treatment modalities for tubal pregnancy to guide optimal therapeutic decision-making. This meta-analysis reviews the existing literature comparing failure rates

of methotrexate (MTX), salpingostomy, and salpingostomy combined with MTX in the management of tubal pregnancy, aiming to evaluate their prognostic outcomes and inform clinical practice.

## 1.1 Inclusion and exclusion criteria

Inclusion criteria: 1. Study designs: Randomized controlled trials (RCTs), observational studies, comparative studies, clinical trials, multicenter studies, or controlled studies. 2. Study population: Women with hemodynamically stable diagnosed tubal pregnancy who have fertility needs, the surgical method is laparoscopic surgery. 3. Interventions: Treatment with methotrexate (MTX), salpingostomy, and salpingostomy combined with methotrexate. In salpingostomy combined with methotrexate, methotrexate is used after surgery. 4. Outcome measures: Inclusion of treatment failure rate as a primary outcome.

Exclusion criteria: 1. Non-tubal ectopic pregnancies (e.g., ovarian or cervical pregnancies). 2. Ultrasonographic evidence of tubal rupture or hemoperitoneum. 3. Animal studies, cell studies, reviews, meta-analyses, case reports, letters, or duplicate publications. 4. Studies with methodological errors, or outcomes inconsistent with inclusion criteria.

## 1.2 Search strategy

A comprehensive search was conducted in PubMed, EMbase, Web of Science, and the Cochrane Library from the inception of these databases to February 2025. Free search terms included "tubal pregnancy and salpingostomy," and "ectopic pregnancy and salpingostomy". The searched literature was all the literature in the four databases that met the inclusion criteria, without restriction of language. Two researchers independently performed the literature search and manual screening. Any discrepancies were resolved through consultation with a third researcher.

## 1.3 Data extraction

Two investigators (Yong Lin and Junwei Liao) independently screened and extracted data from eligible studies. Any disagreements were adjudicated by a third author (Yan Long). Extracted data included: 1. Study characteristics: First author, publication year, study design, sample size, and participant age. 2. Interventions: Methotrexate (MTX), salpingostomy, or combined therapy. 3. Clinical parameters: Baseline serum hCG levels, gestational sac size, and duration of follow-up. 4. Outcomes: Treatment failure rate.

Methotrexate treatment failure is defined as a reduction of less than 15% in the serum human chorionic gonadotropin (HCG) levels on days 4 and 7 following methotrexate administration in cases of tubal pregnancy. Failure of laparoscopic surgery refers to an increase or no decrease in blood HCG on the fourth day after the operation.

## 1.5 Assessment of risk of bias

The risk of bias was evaluated using the Newcastle-Ottawa Scale (NOS) for controlled studies and the modified Jadad scale for RCTs. NOS assesses three domains: selection (0–4 points), comparability (0–2 points), and outcome/exposure (0–3 points). Total scores categorize studies as low (0–3), moderate (4–6), or high quality (7–9) [16]. Modified Jadad scale evaluates randomized controlled trials (RCTs) on a 7-point scale (low quality: 1–3; high quality: 4–7) [17]. Missing data were requested from the corresponding authors. Any discrepancies were resolved via consensus or third researcher. In Jadad scale and NOS, low quality studies have shown a high risk of bias.

## 1.6 Statistical analysis

**1.6.1 Meta-analysis.** Analyses were performed using Stata 17.0. Odds ratios (ORs) with 95% confidence intervals (CIs) were calculated. Heterogeneity was assessed via Cochran's Q test and $I^2$ statistics (significant if $p < 0.1$ or $I^2 > 50\%$). Publication bias was evaluated using funnel plots, Begg's test, and Egger's test regression ($p < 0.05$ indicating bias). Sensitivity analyses and subgroup analyses explored heterogeneity sources.

**1.6.2 Network Meta-Analysis (NMA).** NMA compared interventions using ORs (95% CIs). Consistency between direct and indirect evidence was tested via global and local inconsistency models. Node-splitting analysis identified local discrepancies (p < 0.05). Treatment rankings were derived from SUCRA values (Surface Under the Cumulative Ranking, 0–100%), with higher values indicating superior efficacy [18].

## 1.7 Evaluation of the quality of evidence process

Two researchers independently used the GRADE approach to evaluate the certainty of the evidence. For randomized controlled trials, we evaluated the quality of evidence for each outcome based on the risk of bias, inconsistency, indirectness, publication bias, and imprecision. For observational studies, we evaluated the quality of evidence for each outcome based on effect size, dose-response relationship, and negative offset. The evaluation was performed independently by two researchers. In case of disagreements, a third reviewer was consulted to reach a consensus. The quality of evidence was classified into four levels: high, moderate, low, and very low. High-quality evidence was defined as evidence from RCTs (without factors that could cause the quality of evidence to be downgraded) or observational studies that were upgraded in quality by two levels. Moderate-quality evidence is defined as evidence from RCTs that were downgraded by one level or observational studies that were upgraded by one level. Low-quality evidence was defined as evidence from RCTs that were downgraded by two levels or observational studies. Very low-quality evidence is defined as evidence from RCTs that were downgraded by three levels, observational studies that were downgraded by one level, case series, and case reports [19,20].

## 1.8 Ethical approval

All analyses were based on published studies and did not require ethical approval or patient consent.

## 2 Results

### 2.1 Literature search process and characteristics of the studies

An initial search yielded 609 articles. After reviewing the titles and abstracts, 19 articles remained. Following the removal of duplicates and a comprehensive full-text review, 8 studies were ultimately included in the meta-analysis [21–28]. The literature search process is shown in Fig 1. The basic characteristics of the included studies are shown in Table 1.

### 2.2 Risk of bias assessment of included studies

The Jadad scale and NOS were used to evaluate the risk of bias in the included studies. The Jadad scale was used to evaluate the quality of randomized controlled trials. The quality scores of the relevant articles ranged from 4 to 6 points. The NOS was used to evaluate the quality of case-control studies, with the quality scores of the relevant studies being 6 points. All studies were of high quality. The quality scores of the studies are shown in Table 2.

### 2.3 Assessment of the level of evidence quality

**2.3.1 Failure rate of salpingostomy combined with methotrexate compared to salpingostomy alone for tubal pregnancy.** The meta-analysis comprised two studies: one controlled trial and one randomized controlled trial (RCT). The effect size of the controlled trial was small (odds ratio [OR] = 0.01) and did not present any significant confounders; therefore, the quality of evidence improved by one level. The quality of evidence in randomized controlled trial remained unchanged after evaluation. Consequently, the overall level of evidence quality for this meta-analysis ranges from intermediate to high. The results are shown in S1 Table.

**2.3.2 Failure rate of methotrexate compared to salpingostomy for tubal pregnancy.** The meta-analysis included six studies: one controlled trial and five RCTs. Among the control trial, despite not having any significant confounding

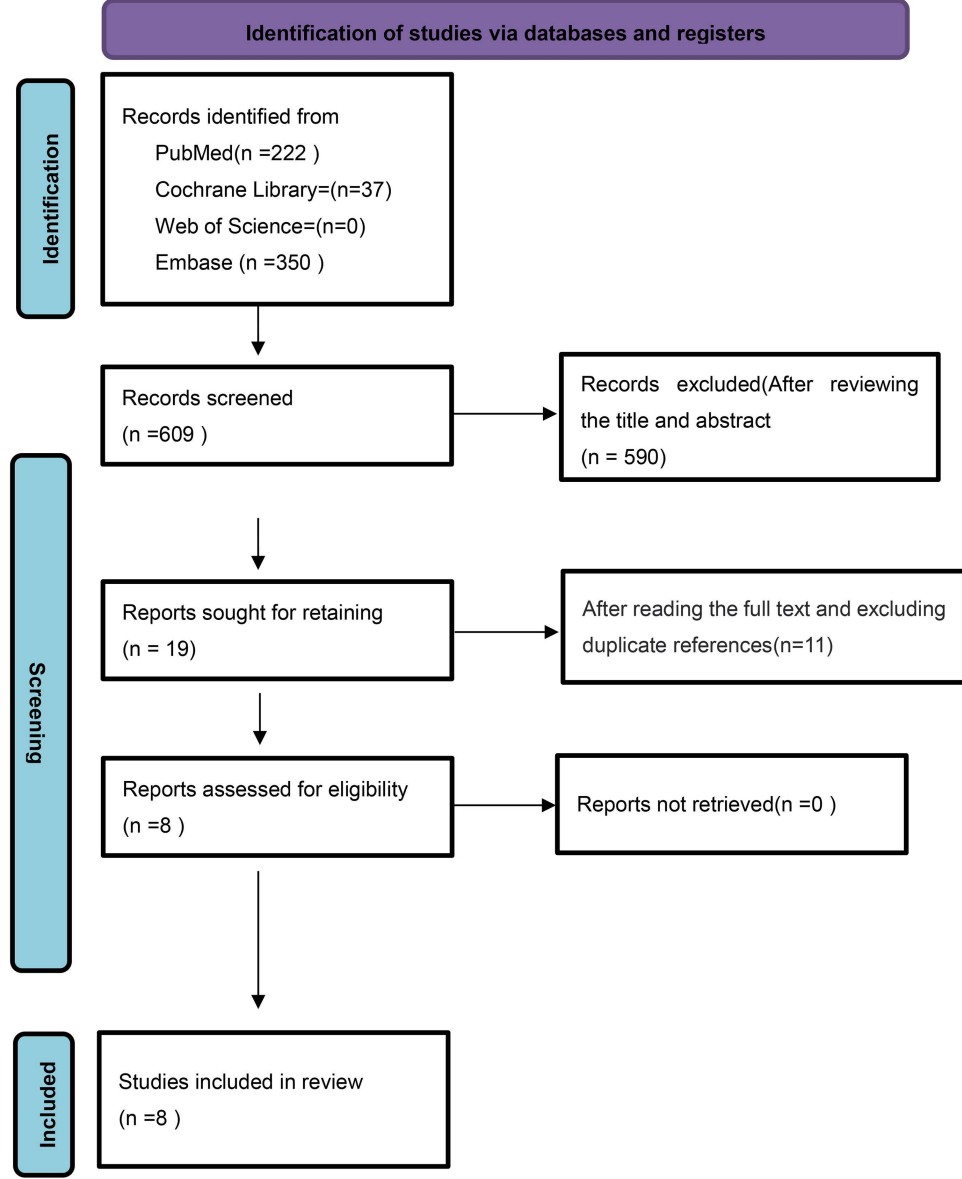

**Fig 1. The retrieval process of the studies.**

factors, the effect size was modest, and the quality of evidence did not improve. The quality of evidence in the randomized controlled trials did not downgrade after evaluation. Therefore, the overall level of evidence quality for this meta-analysis ranges from low to high. The results are shown in S1 Table.

### 2.4 Meta-analysis results

**2.4.1 Failure rate of salpingostomy combined with methotrexate compared to salpingostomy alone for tubal pregnancy.** This analysis included 3 studies involving 262 participants. A common inverse-variance model was used for the meta-analysis to aggregate the odds ratios (ORs) from each study. The meta-analysis revealed an overall OR

**Table 1. The basic characteristics of the included studies.**

| Study | geographic region | ethnicity | Study type | age | | Sample | | Gestational sac diameter(cm) | | average HCG | |
|---|---|---|---|---|---|---|---|---|---|---|---|
| Salpingostomy combined with MTX/MTX | | | | | | | | | | | |
| J W Graczykowski 1997 | Los Angeles, USA | – | RCT | 27.9±5.3 | 26.2±4.4 | 54 | 62 | – | – | – | – |
| Shigeo Akira 2008 | Tokyo | – | controlled trial | 29.2±5.3 | 29.8±4.0 | 41 | 40 | – | – | 3287±3230 (250-13012) | 3165±2758 (198-12756) |
| Hakan Kaya, M 2002 | Isparta,Turkey | – | RCT | 28.8±5.2 | 30.1±5.8 | 22 | 43 | 2.4±0.9 | 2.3±1.0 | 245-7800 | 450-7200 |
| Methotrexate/salpingostomy | | | | | | | | | | | |
| P J Hajenius 1997 | Amsterdam, Netherlands | – | RCT | 31.3±5.9 | 31.8±4.4 | 51 | 49 | – | – | 1950(110-19500) | 2100(228-18400) |
| F.Lecuru 1998 | Paris, France | – | controlled trial | 31 (17-45) | 29 (19-38) | 37 | 38 | – | – | 585(50-5525) | 998(98-6000) |
| ASMA J. SARAJ 1998 | Los Angeles, USA | – | RCT | 28.1±1.2 | 27.4±1.0 | 38 | 36 | 2.7±0.1 | 2.7±0.3 | 3162±772 | 3357±766 |
| Martin C. Sowter 2001 | Auckland, New Zealand | | RCT | 29.7±5.4 | 30.4±4.6 | 34 | 26 | – | – | 927 (137 - 4866) | 775 (89 - 4800) |
| Lars Bo Krag Moeller 2009 | Hvidovre, Denmark | – | RCT | 30.8 (21.1–40.9) | 32.7 (24.3–41.2) | 53 | 53 | 1.2 (0.5-2.5) | 1.4 (0.7-2.9) | 2259 (176-4100) | 3200 (72-42859) |

**Table 2. Quality scores of the included studies.**

**A NOS**

| Study | I | II | III | IV | V | VI | VII | VIII | Total points |
|---|---|---|---|---|---|---|---|---|---|
| Shigeo Akira 2008 | 1 | 0 | 1 | 1 | 1 | 1 | 1 | 0 | 6 |
| F.Lecuru 1998 | 1 | 0 | 1 | 1 | 1 | 1 | 1 | 0 | 6 |

**B Jadad score**

| Study | Generation of the random sequences | Randomization hidden | blind method | Withdrawal and loss of follow-up | total points |
|---|---|---|---|---|---|
| J W Graczykowski 1997 | 2 | 1 | 1 | 1 | 5 |
| Martin C. Sowter 2001 | 2 | 2 | 1 | 1 | 6 |
| Hakan Kaya, M 2002 | 1 | 1 | 1 | 1 | 4 |
| P J Hajenius 1997 | 2 | 2 | 1 | 1 | 6 |
| ASMA J. SARAJ 1998 | 2 | 2 | 1 | 1 | 6 |
| Lars Bo Krag Moeller 2009 | 2 | 2 | 1 | 1 | 6 |

Note:I Case identif-ication is appropriate. II Repres-entative of cases. III The selection of controls. IV Determ-ination of contrast. V The comparability of cases and controls was considered for the design and statistical analysis. VI Determi-nation of the exposure factors VII The same method was used to determine the exposure factors in the cases and controls. VIII No response rate.

of 0.11 (95% CI: 0.03–0.48), indicating a significant overall effect (p = 0.003). The heterogeneity among the studies was low (I² = 0.0%), indicating high consistency across the studies. These results suggest that salpingostomy combined with methotrexate significantly reduces the failure rate of treating tubal pregnancies compared to salpingostomy alone.This result is shown in Fig 2.

**2.4.2 Failure rate of methotrexate compared to salpingostomy for tubal pregnancy.** This analysis included 5 studies involving 415 participants. A fixed-effects inverse-variance model was used for the meta-analysis to aggregate the

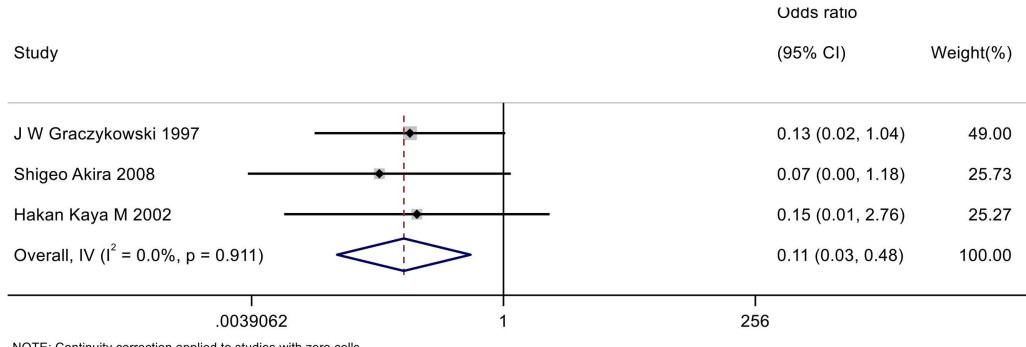

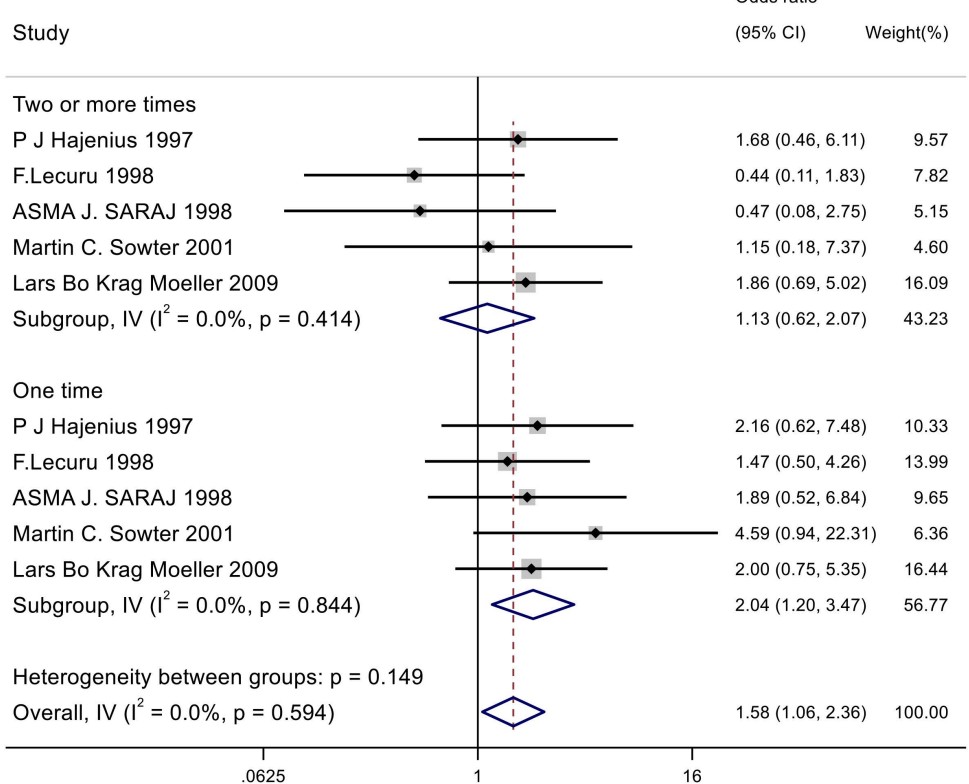

**Fig 2. The forest plot of failure rate of salpingostomy combined with methotrexate compared to salpingostomy alone for tubal pregnancy.**

ORs from each study. The studies were categorized based on the number of methotrexate interventions ("single dose" and "two or more doses"), and subgroup analyses were performed.

The meta-analysis showed an overall OR of 1.582 (95% confidence interval [CI]: 1.062–2.358), with a significant overall effect (p = 0.024). Subgroup analysis demonstrated that a single dose of methotrexate was associated with a higher failure rate compared to salpingostomy(OR = 2.044, p = 0.008). In contrast, two or more doses of methotrexate did not show a significant difference in failure rate compared to salpingostomy(OR = 1.130, p = 0.692). The heterogeneity among the studies was low (I²=0.0%), suggesting a high level of consistency across the studies. This result is shown in Fig 3.

**Fig 3. The forest plot of failure rate of methotrexate compared to salpingostomyfor tubal pregnancy.**

## 2.5 Network meta-analysis results

### 2.5.1 Comparison of failure rates among three treatment modalities for tubal pregnancy.

This analysis included 8 studies involving 677 participants. We compared the failure rates of three treatment modalities: methotrexate (two or more doses), salpingostomy combined with methotrexate, and salpingostomy alone, using a network meta-analysis. The results are as follows: Methotrexate versus salpingostomy combined with methotrexate: The ORs were −2.46 and 2.46, respectively. The confidence intervals did not include 0, indicating a significant difference in efficacy between the two treatments, with salpingostomy combined with methotrexate demonstrating greater effectiveness. Methotrexate versus salpingostomy: The ORs were −0.11 and 0.11, respectively, and the confidence intervals included 0, indicating no significant difference in efficacy between the two treatments. Salpingostomy combined with methotrexate versus salpingostomy: The effect sizes were 2.35 and −2.35, respectively, and the confidence intervals did not include 0, indicating a significant difference in efficacy. This suggests that salpingostomy combined with methotrexate is more effective. This result is shown in Table 3.

SUCRA analysis indicated that salpingostomy combined with methotrexate is the most effective potential intervention, and has lowest failure rate. This result is shown in Fig 4.

**Table 3. The comparative results of network meta-analysis.**

| methotrexate | salpingostomy combined with methotrexate | salpingostomy |
|---|---|---|
| methotrexate | −2.46 (−4.15,-0.77) | −0.11 (−0.84,0.62) |
| 2.46 (0.77,4.15) | salpingostomy combined with methotrexate | 2.35 (0.82,3.88) |
| 0.11 (−0.62,0.84) | −2.35 (−3.88,-0.82) | salpingostomy |

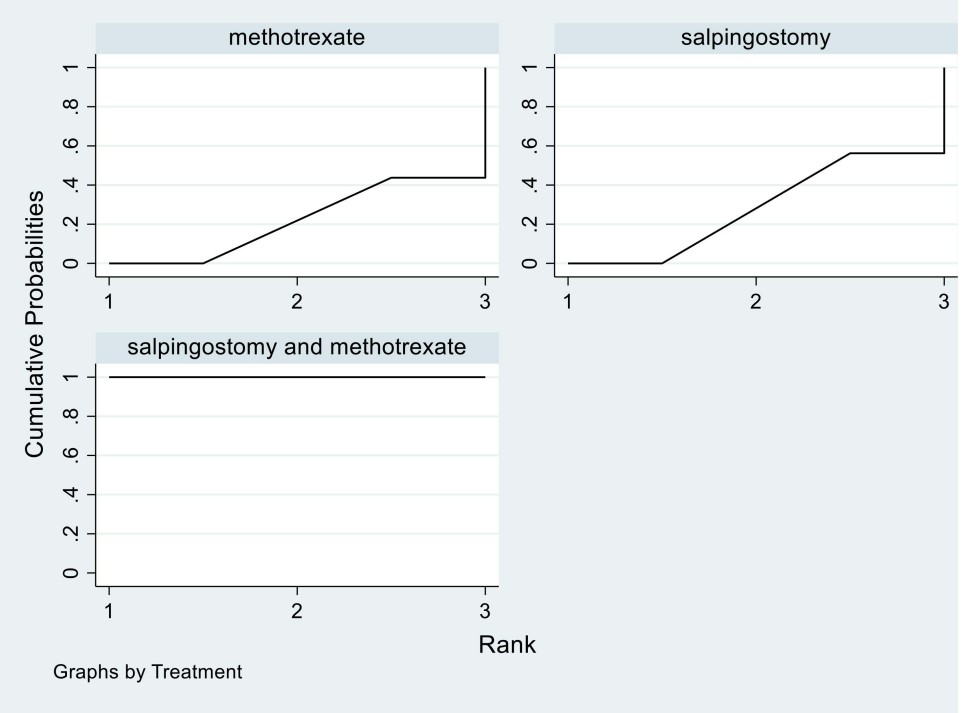

**Fig 4. The SUCRA results of three treatments.**

## 2.6 Publication bias

The funnel plots of the included studies appeared to be roughly symmetrical. Additionally, we conducted Begg's test and Egger's test to assess the presence of publication bias in this study, and no significant publication bias was detected. This result is shown in Figs 5 and 6.

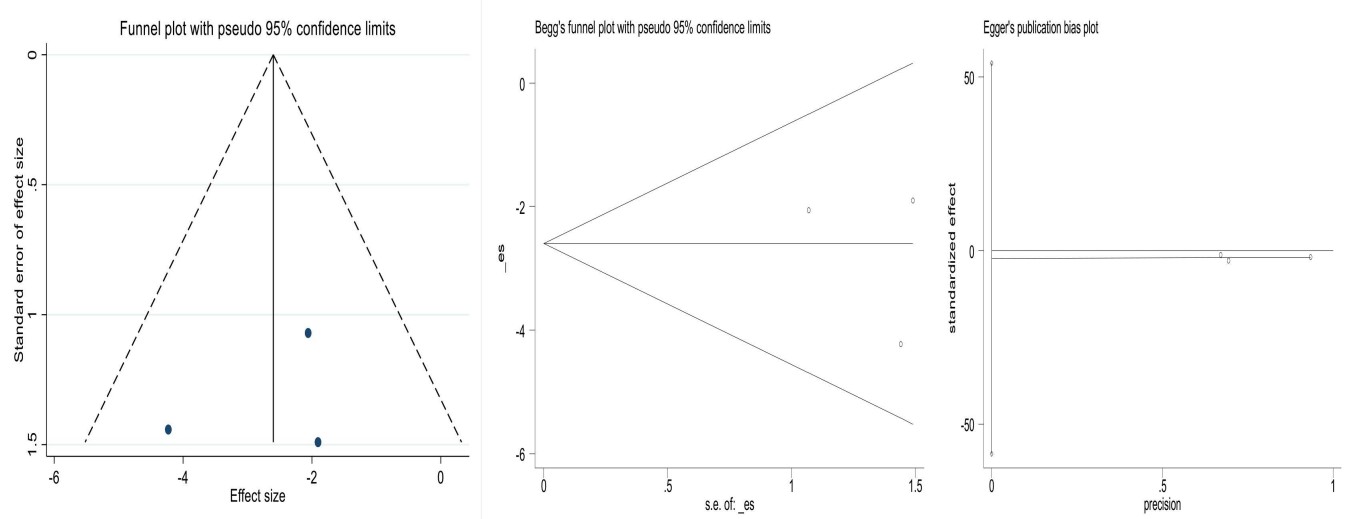

**Fig 5. Funnel plot of failure rate of salpingostomycombined with methotrexate compared to salpingostomyalone for tubal pregnancy (Begg's Test p = 1.000, Egger's Test p = 0.700).**

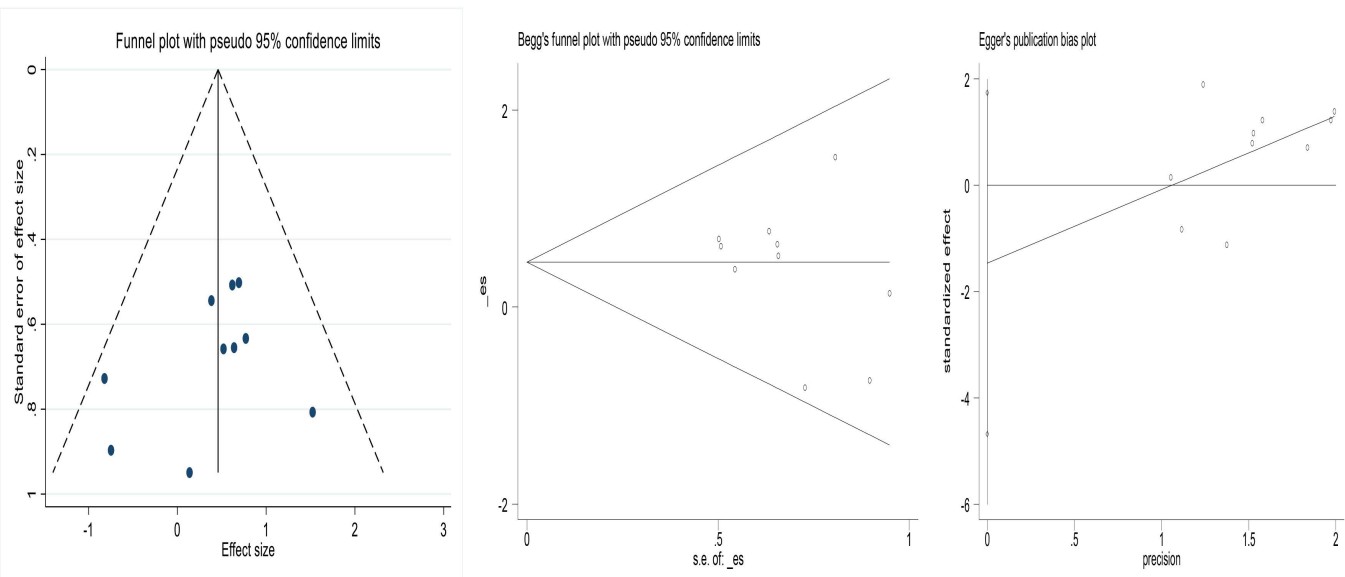

**Fig 6. Funnel plot of failure rate of methotrexate compared to salpingostomyfor tubal pregnancy(Begg's Test p = 1.000, Egger's Test p = 0.204).**

## 2.7 Sensitivity analysis

To evaluate the robustness of the results and examine heterogeneity, sensitivity analyses were conducted by excluding each study individually. The findings remained consistent even when any single study was omitted in the sensitivity analysis. The results of the sensitivity analysis are shown in Fig 7.

## 2.8 Inconsistency test

In the current study, the network evidence diagram we constructed exhibited significant flaws, as its structure did not form a closed loop. This limitation hindered our ability to conduct overall and local inconsistency tests based on the evidence diagram. Consequently, no inconsistency tests were performed. The results are shown in Fig 8.

## 3 Discussion

Salpingostomy, a fertility-preserving surgical approach, plays a critical role in the management of tubal pregnancies by preserving the affected fallopian tube, thereby maintaining the potential for future natural conception. This approach is particularly vital for patients desiring fertility, especially those with contralateral tubal dysfunction or absence. Conversely, other studies reported no significant differences in intrauterine pregnancy (IUP) or repeat ectopic pregnancy (REP) rates between salpingostomy and methotrexate (MTX) therapy [6].

MTX, a folate antagonist, terminates pregnancy by inhibiting cellular proliferation and disrupting trophoblastic cells. Single-dose MTX regimens are widely recognized as safe and effective for hemodynamically stable patients with early-diagnosed ectopic pregnancies [6].

However, our analysis shows that in general, the failure rate of methotrexate (MTX) is higher than that of salpingostomy; while the subgroup analysis indicates that the failure rate of single-dose MTX treatment is higher than that of salpingostomy, and the failure rate of the double-dose MTX regimen does not show a significant difference compared to

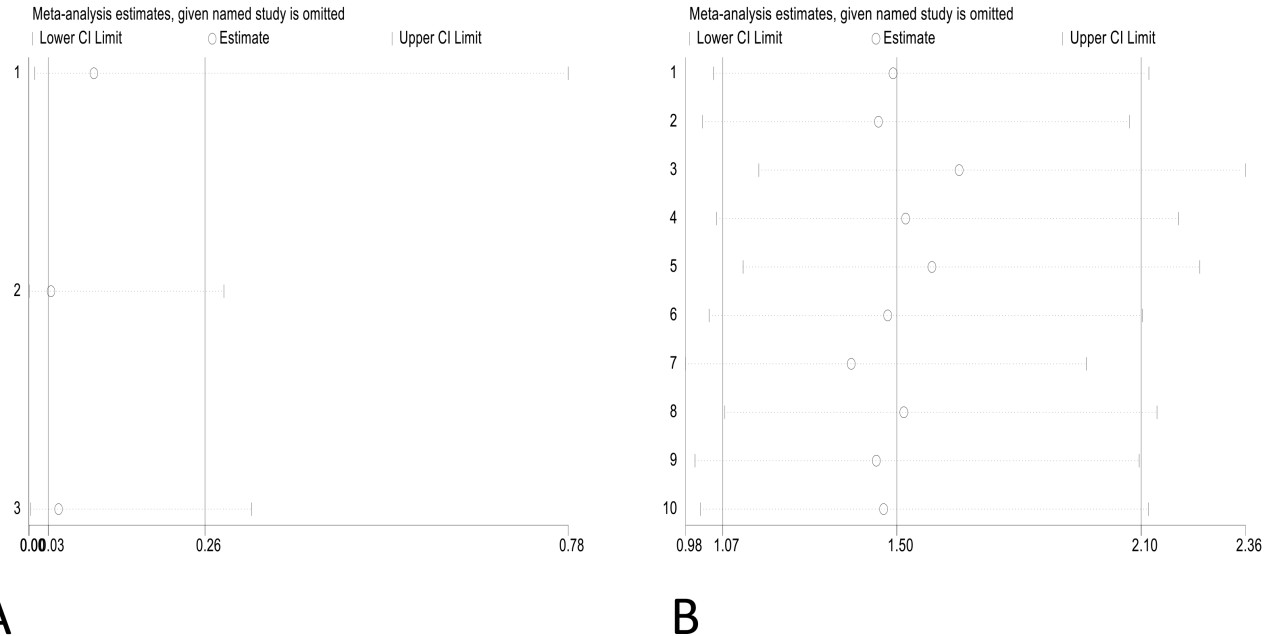

**Fig 7. Sensitivity analysis.** A Failure rate of salpingostomycombined with methotrexate compared to salpingostomyalone for tubal pregnancy. B Failure rate of methotrexate compared to salpingostomyfor tubal pregnancy.

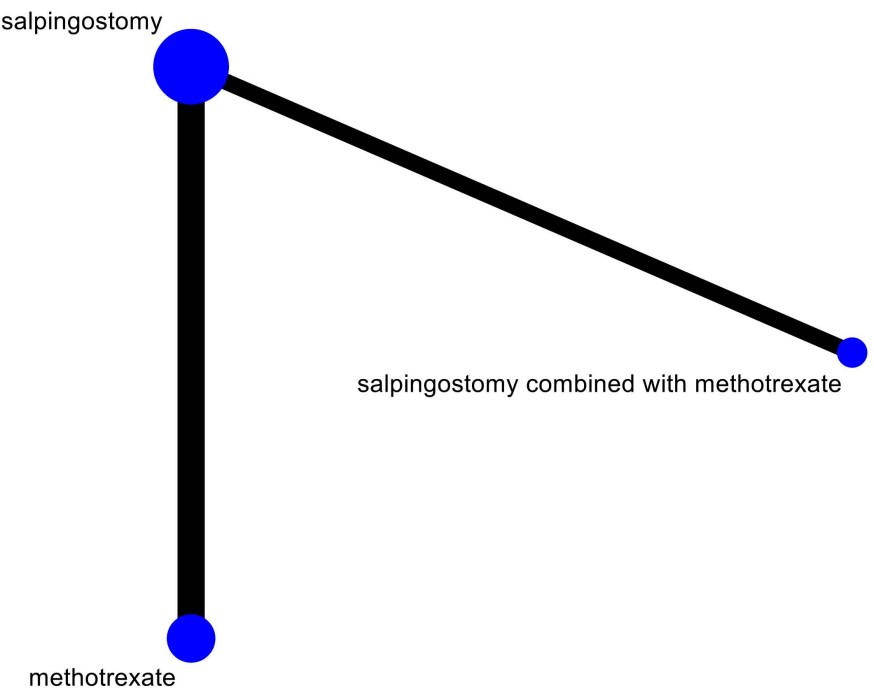

**Fig 8. Network evidence diagram.**

salpingostomy. These suggest that a single-dose intervention with methotrexate may be associated with a higher risk of treatment failure, while two doses administrations do not show a significant difference, indicating that in clinical practice, it may be necessary to optimize the methotrexate administration regimen (such as increasing the dose) to reduce the failure rate, but this needs further verification.

Our meta-analysis shows also that the failure rate of salpingostomy combined with methotrexate treatment for tubal pregnancy is significantly lower than that of simple salpingostomy. Through network meta-analysis, it is further confirmed that salpingostomy combined with MTX treatment is the most effective, and compared with methotrexate and simple salpingostomy, salpingostomy combined with MTX has a lower failure rate while preserving the function of the fallopian tube.

According to our meta-analysis results, when hemodynamics were stable, we observed that the levels of blood HCG on days 4 and 7 following the first treatment with methotrexate were unsatisfactory, that is, the decrease in blood HCG levels is less than 15%. If hemodynamics were still stable, we could inject methotrexate again to treat the tubal pregnancy.

Cost-effectiveness analyses indicate that laparoscopic salpingostomy remains the most cost-effective approach. Multi-dose MTX is considered cost-effective only when serum human chorionic gonadotropin (hCG) concentrations are below 3,000 IU/L, whereas single-dose MTX becomes cost-effective at hCG levels below 1,500 IU/L [29]. For patients with initial hCG levels between 1,500 and 3,000 IU/L, the costs of salpingostomy and MTX are comparable; however, MTX becomes more expensive when hCG levels exceed 3,000 IU/L [30].

Health-related quality of life (HRQoL) studies indicate that systemic methotrexate (MTX) therapy is linked to greater impairments in physical, role, and social functioning, as well as poorer health perception, increased pain, and higher rates of depression compared to surgical management [31]. Though salpingostomy combined with MTX reduces failure rates, multimodal therapy may be result in more side effects.

## 4 Strengths and limitations

A key strength of this study is the integration of conventional meta-analysis and network meta-analysis to evaluate treatment outcomes for tubal pregnancies. The inclusion of randomized controlled trials (RCTs) across diverse clinical settings enhances the generalizability of the findings to real-world scenarios. The pooled effects in standard meta-analyses demonstrated robust statistical power, and sensitivity analyses confirmed the stability of the results.

However, there are several limitations, including: 1. A limited number of randomized controlled trials (RCTs) necessitates the conduct of higher-quality studies to strengthen the conclusions. 2. Small sample sizes in some of the included studies may potentially amplify publication bias.

## 5 Conclusion

For hemodynamically stable patients with a desire for future fertility, salpingostomy combined with methotrexate is an excellent option for treating tubal pregnancy. This approach has a lower failure rate compared to either methotrexate or salpingostomy alone and appears to have minimal impact on the patient's quality of life.

## Supporting information

**S1 Table. Results of the evidence quality assessment of the studies.**
(XLSX)

## Acknowledgments

The authors would like to express their gratitude to all the researchers in our research group.

## Author contributions

**Writing – original draft:** Yong Lin, Junwei Liao.

**Writing – review & editing:** Yong Lin, Junwei Liao, Yan Long.

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
