## [Decision Letter · Decision Letter 0]

16 Jul 2025

PONE-D-25-17065Comparison of tubal-preserving surgical management versus methotrexate therapy for tubal pregnancy: a conventional and network meta-analysisPLOS ONE

Dear Dr. Lin,

Thank you for submitting your manuscript to PLOS ONE. After careful consideration, we feel that it has merit but does not fully meet PLOS ONE’s publication criteria as it currently stands. Therefore, we invite you to submit a revised version of the manuscript that addresses the points raised during the review process.

We look forward to receiving your revised manuscript.

Kind regards,

Li Zhe

Academic Editor

PLOS ONE

Journal Requirements:

The authors claim there is no conflict of interest

3. Please ensure that you refer to Figures 5 and 6 in your text as, if accepted, production will need this reference to link the reader to the figures.

4. Please remove all personal information, ensure that the data shared are in accordance with participant consent, and re-upload a fully anonymized data set.

Additional guidance on preparing raw data for publication can be found in our Data Policy (https://journals.plos.org/plosone/s/data-availability#loc-human-research-participant-data-and-other-sensitive-data) and in the following article: http://www.bmj.com/content/340/bmj.c181.long .

Additional Editor Comments:

This munuscript is quite interesting, but there are some issues that must be addressed.

First, regarding study inclusion, in addition to the factors already listed by the authors, geographic region and ethnicity are also critically important, as they directly impact treatment effectiveness.

Second, the figures and tables in the article suffer from inconsistent font styles and cluttered formatting, which need to be improved for better readability and presentation.

Third, in the Discussion section, much of the current content would be more appropriately placed in the Introduction. The discussion should instead focus on your findings and their interpretation.

Although the two reviewers did not raise many concerns, I still strongly recommend that this manuscript undergo thorough revision.

Reviewers' comments:

Reviewer's Responses to Questions

**Comments to the Author**

1. Is the manuscript technically sound, and do the data support the conclusions?

Reviewer #1: Yes

Reviewer #2: Yes

2. Has the statistical analysis been performed appropriately and rigorously? 

Reviewer #1: Yes

Reviewer #2: Yes

3. Have the authors made all data underlying the findings in their manuscript fully available?

Reviewer #1: Yes

Reviewer #2: Yes

4. Is the manuscript presented in an intelligible fashion and written in standard English?

Reviewer #1: Yes

Reviewer #2: No

5. Review Comments to the Author

Reviewer #1: I thank the author for taking out time to carryout this very important piece of work.

There are few queries I would want the authors to adress

1. There are continuing concerns with recurrent ectopic pregnancy following salpingotomy/salpingostomy due to scarring. Kindly address this.

2. In selecting the works on medical management of tubal pregnancy, did you consider the size of the ectopic mass? If yes, what was the eligible maximum diameter for medical management. This is very important as this, in addition to the level of hCG, is a determinant of success rate.

3. Justify your use of decrease of less than 20% on days 4 & 7 as your definition of failure of medical management. A change of <15% between the day 4 and 7 values is widely used and we know that it is usual not to have a significant decrease or even an increase level on day 4 level following medical management.

4. You appear to interchange the use salpingotomy and salpingostomy eventhough they are different procedures. Can you rectify this please.

Reviewer #2: While the manuscript generally reflects scientific research that supports its conclusions, some parts of the discussion—particularly those addressing quality of life and cost—lack accompanying data analysis. The conclusions in these sections seem to be derived from studies conducted by other researchers.

6. PLOS authors have the option to publish the peer review history of their article (what does this mean? ). If published, this will include your full peer review and any attached files.

**Do you want your identity to be public for this peer review?** For information about this choice, including consent withdrawal, please see our Privacy Policy .

Reviewer #1: No

Reviewer #2: No

---

## [Decision Letter · Decision Letter 1]

19 Aug 2025

PONE-D-25-17065R1Comparison of tubal-preserving surgical management versus methotrexate therapy for tubal pregnancy: a conventional and network meta-analysisPLOS ONE

Dear Dr. Lin,

Thank you for submitting your manuscript to PLOS ONE. After careful consideration, we feel that it has merit but does not fully meet PLOS ONE’s publication criteria as it currently stands. Therefore, we invite you to submit a revised version of the manuscript that addresses the points raised during the review process.

**Please reviese the manuscript depend on Reviewer2's comment.**

We look forward to receiving your revised manuscript.

Kind regards,

Li Zhe

Academic Editor

PLOS ONE

Journal Requirements:

Reviewers' comments:

Reviewer's Responses to Questions

**Comments to the Author**

1. If the authors have adequately addressed your comments raised in a previous round of review and you feel that this manuscript is now acceptable for publication, you may indicate that here to bypass the “Comments to the Author” section, enter your conflict of interest statement in the “Confidential to Editor” section, and submit your "Accept" recommendation.

Reviewer #1: All comments have been addressed

Reviewer #2: All comments have been addressed

2. Is the manuscript technically sound, and do the data support the conclusions?

Reviewer #1: Yes

Reviewer #2: Yes

3. Has the statistical analysis been performed appropriately and rigorously? 

Reviewer #1: Yes

Reviewer #2: Yes

4. Have the authors made all data underlying the findings in their manuscript fully available?

Reviewer #1: Yes

Reviewer #2: Yes

5. Is the manuscript presented in an intelligible fashion and written in standard English?

Reviewer #1: Yes

Reviewer #2: Yes

6. Review Comments to the Author

Reviewer #1: I wish to thank the authors for taking out the time to address the queries. I have no further concerns.

Reviewer #2: Considering that not all hospitals have adequate resources and facilities, particularly for laparoscopic procedures, it would be advisable to address this aspect in the manuscript. While this may not substantially alter or diminish the overall value of your publication, it is nonetheless a point worth noting.

7. PLOS authors have the option to publish the peer review history of their article (what does this mean? ). If published, this will include your full peer review and any attached files.

**Do you want your identity to be public for this peer review?** For information about this choice, including consent withdrawal, please see our Privacy Policy .

Reviewer #1: No

Reviewer #2: No

---

## [Editor Report · Decision Letter 2]

27 Aug 2025

Comparison of laparoscopic tubal-preserving surgical management versus methotrexate therapy for tubal pregnancy: a conventional and network meta-analysis

PONE-D-25-17065R2

Dear Dr. Lin,

We’re pleased to inform you that your manuscript has been judged scientifically suitable for publication and will be formally accepted for publication once it meets all outstanding technical requirements.

Kind regards,

Li Zhe

Academic Editor

PLOS ONE
---

## [Editor Report · Acceptance letter]

PONE-D-25-17065R2

PLOS ONE

Dear Dr. Lin,

I'm pleased to inform you that your manuscript has been deemed suitable for publication in PLOS ONE. Congratulations! Your manuscript is now being handed over to our production team.

Kind regards,

on behalf of

Dr. Li Zhe

Academic Editor

PLOS ONE